# Recent Advances in Metal–Organic Framework-Based Anticancer Hydrogels

**DOI:** 10.3390/gels11010076

**Published:** 2025-01-18

**Authors:** Preeti Kush, Ranjit Singh, Parveen Kumar

**Affiliations:** 1Adarsh Vijendra Institute of Pharmaceutical Sciences, Shobhit University Gangoh, Saharanpur 247341, Uttar Pradesh, India; ranjit.singh@shobhituniversity.ac.in; 2Exigo Recycling Pvt Ltd., Karnal 132114, Haryana, India

**Keywords:** metal–organic framework, hydrogel, chemotherapy, challenges

## Abstract

Cancer is the second leading cause of death globally and the estimated number of new cancer cases and deaths will be ∼30.2 million and 16.3 million, respectively, by 2040. These numbers cause massive, physical, emotional, and financial burdens to society and the healthcare system that lead to further research for a better and more effective therapeutic strategy to manage cancer. Metal–organic frameworks (MOFs) are promising alternative approaches for efficient drug delivery and cancer theranostics owing to their unique properties and the direct transportation of drugs into cells followed by controlled release, but they suffer from certain limitations like rigidity, poor dispersibility, fragility, aggregation probability, and limited surface accessibility. Therefore, MOFs were conjugated with polymeric hydrogels, leading to the formation of MOF-based hydrogels with abundant absorption sites, flexibility, and excellent mechanical properties. This review briefly describes the different strategies used for the synthesis and characterization of MOF-based hydrogels. Further, we place special emphasis on the recent advances in MOF-based hydrogels used to manage different cancers. Finally, we conclude the challenges and future perspectives of MOF-based hydrogels. We believe that this review will help researchers to develop more MOF-based hydrogels with augmented anticancer effects, enabling the effective management of cancer even without adverse effects.

## 1. Introduction

Cancer is the second leading cause of death globally and affects approximately 60 organs [1]. As per the World Health Organization (WHO), the estimated number of deaths and estimated new cancer cases will be ∼16.3 million and 30.2 million, respectively, by 2040 [2]. These astonishing numbers cause massive, physical, emotional, and financial burdens to the individuals, families, the healthcare system, and communities that lead to further research for a better and more effective therapeutic strategy to manage cancer [3]. According to the WHO, cancer can be effectively managed by taking preventive measures, early detection, and using a suitable therapeutic regimen. Early detection comprises the preliminary diagnosis to recognize symptomatic cancer at the initial stage and screening to recognize pre-cancer patients who have not any abnormalities or symptoms related to cancer and refer them for further detection and treatment. Cancer is managed by various therapeutic strategies like radiotherapy, immunotherapy [4,5], surgery [6], chemotherapy [7], hormonal therapy [8], laser therapy [9], stem cell transplant, and targeted therapies [10]. Additionally, other novel strategies such as natural antioxidants [11], nano catalytic therapy [chemodynamic therapy (CDT), electrodynamic therapy (EDT), electrochemical therapy (EChT), microwave dynamic therapy (MDT), magnetic hyperthermia (MH), photodynamic therapy (PDT), photothermal therapy (PTT), pyroelectric-dynamic therapy (PEDT), sonodynamic therapy (SDT), thermodynamic therapy (TDT)], ablation therapy (cryo- and thermal ablation), gene therapy, hyperthermia, and starvation therapy (ST) have been also explored to manage cancer. The selection of the therapeutic strategy depends on the type of cancer, its stage, and clinical factors; therefore, these can be either used alone or in combination [2]. Among all, chemotherapy is the most preferential clinical strategy but suffers from various limitations including non-specific targeting, poor accessibility to the tumor tissues, toxicity to normal cells, and drug resistance. Therefore, various nanoformulations including metallic or polymeric nanoparticles (NPs), lipid-based nanoformulations (cubosomes, double emulsions, ethosomes, liposomes, phytosomes, solid lipid NPs, and transferosomes), protein-based (e.g., albumin, gelatin, milk, polypeptide, sericin, zein proteins) and carbohydrate-based (e.g., cyclodextrins, chitosan, other polysaccharides) nanoformulations, and multi-biopolymers systems have been explored proposed to overcome such limitations [3]. However, these nanoformulations have different concerns like poor drug loading, burst release, biocompatibility, and stability [12].

Metal–organic frameworks (MOFs) are hybrid porous organic–inorganic coordination compounds composed of organic linkers (e.g., carboxylates, imidazolates, phosphonates, etc.) and metal ions (e.g., Ag, Fe, Cr, Co, Cu, Ni, Zn, etc.). These nanocarriers are promising alternative approaches for efficient drug delivery and cancer theranostics owing to their unique properties like their amphiphilic microenvironment, adaptable shape and pore size, diverse morphology, adjustable functionalities, intrinsic biocompatibility and biodegradability, stimuli sensitivity, and large surface-area-to-volume ratio. Moreover, MOFs can transport the drug into cells and release the drug in a controlled manner [13,14,15,16,17]. Further, MOFs can be used for different biomedical applications (e.g., biosensing, enantioselective recognition, imaging) [16,18], catalysis [19], separation and purification [20], and the storage of energy [21] due to their unique properties. Despite their various distinct properties, MOFs are not clinically used in drug delivery applications due to certain limitations like their rigidity, perturbed processability [22], poor dispersibility, and aggregation probability [23]. Moreover, MOFs exhibited inherent fragility during exposure to an aqueous environment leading to the decomposition of the MOF structure [20]. Further, MOFs may be toxic and their toxicity is significantly affected by their composition, size, dosage, and stability [23]. Additionally, MOFs are generally synthesized in crystalline powder form with a large pore structure, but most of them are micropores, leading to limited surface accessibility [22,23]. Therefore, MOFs were conjugated with other matrix or functional materials to achieve more flexibility and stability along with a higher surface area, excellent surface accessibility, and adjustable processability [24]. Various functional materials like ionic liquids, graphene oxide, polyoxometalates, hydrogels, etc., have been used to form MOFs-based composites. Among all the functional materials, hydrogels are the more advantageous functional material to form MOFs-based composite i.e. MOF-based gel owing to their three-dimensional (3-D) structure, large functional groups, flexibility, and abundant absorption sites on the hydrogel network. Moreover, the hydrogel can absorb more water even without losing its structural integrity [20]. Further, hydrogels ensure the uniform distribution of MOF within the polymer matrix bypassing aggregation and can also act as a reservoir for the drug-loaded MOFs to control the release of drugs without toxicity. Contrarily, MOFs can also augment the mechanical properties of the hydrogels, enabling their suitability for biomedical applications Therefore, the distinct properties of both hydrogels and MOFs are complementary and advantageous, making them suitable for synergistic integration [24].

MOF-based hydrogels are widely used in various applications like adsorption, catalysis, energy storage, water treatment, and supercapacitors owing to their unique properties such as magnified porosity, aqueous stability, excellent mechanical properties, large adsorption capacity, biodegradability, tissue mimicry, and biocompatibility [25]. Moreover, these hydrogels are stimuli-responsive, leading to improved drug release in a controlled and progressive manner [24]. These hydrogels are usually prepared by the introduction of MOFs into the polymeric hydrogel matrices by the direct mixing and in situ growth methods [18]. The direct mixing method involves the physical entrapment of MOFs during the cross-linking of the hydrogel matrix, whereas the in situ growth method implies the extension of MOF structures in the pores of a pre-prepared hydrogel. Inside the gel, MOFs are uniformly embedded and distributed [20]. Various natural (chitosan, carrageenan, gelatin, sodium alginate, pectin, etc.) and synthetic polymers (polyvinyl alcohol (PVA), polyethylene glycol (PEG), poly(D,L-lactide-coglycolide) (PLGA), polyacrylic acid, polyacrylamide, etc.) have been used to prepare the hydrogel matrix [23]. There are various articles covering the application of MOF-based hydrogels in different fields. Still, to our best knowledge, there is no review focusing specifically on the application of MOF-based hydrogels for cancer management.

This review briefly covers the synthesis and characterization of MOF-based hydrogel strategies with their pros and cons. Special emphasis has been provided on recent advances in MOF-based hydrogels used to manage different cancers using various strategies, followed by challenges and future perspectives. It is believed that this review will help the researchers to develop more MOF-based hydrogels with augmented anticancer effects, enabling the effective management of cancer even without adverse effects.

## 2. Methodology

The review retrieved papers published in English from Scopus, Google Scholar, PubMed Central, Science Direct, and PubMed using keywords such as “Metal-organic framework, hydrogel, cancer, MOF-based hydrogel, chemotherapy, and carcinoma”. This review includes research and review articles focused on MOF-based anticancer hydrogels from 2017 to 2024. Some information such as Encyclopedia conference proceedings, letters to editors, book chapters, unpublished data, articles published in non-English, and research with limited relevance was excluded.

## 3. MOF-Based Hydrogel: Synthesis and Characterization

MOF-based hydrogel is the composite form of hydrogel and MOFs, possessing the properties of both. These nanocarriers are promising in drug delivery applications in contrast to other nanoformulations, overcoming certain limitations such as poor drug loading, systemic adverse effects, toxicities, and instability owing to the unique properties of MOFs (Figure 1) [26]. Moreover, hydrogel provides large functional groups, flexibility, abundant absorption sites, excellent water retention, and the uniform distribution of MOF within the polymer matrix bypassing aggregation, and it can also act as a reservoir for the drug-loaded MOFs to control the release of drugs without toxicity [24]. This section covers different strategies used for the synthesis of MOF-based hydrogel and their characterization.

### 3.1. Synthesis of MOF-Based Hydrogel

MOF-based hydrogels are usually prepared by introducing MOFs into the polymeric hydrogel matrices. The MOF can be considered a dispersed phase whereas the hydrogel matrix is a continuous phase [22]. These hydrogels are usually prepared by in situ growth, the simultaneous formation of MOF and hydrogel, and the direct mixing method (Figure 2) [20]. Table 1 summarizes the different strategies used for the synthesis of MOF-based hydrogels [20,23,24,27,28].

#### 3.1.1. In Situ Growth Method

The in situ growth method involves the direct formation of MOF into the pores of the hydrogel matrix (Figure 2A). Briefly, the hydrogel matrix is prepared first by the addition of metal precursors into the hydrogel matrix and attached to the hydrogel network; then, ligands are added to interact with the metal ions, resulting in the formation of MOF-based hydrogel (Figure 2A(a)) [20,23]. Alternatively, the in situ growth method involves the addition of hydrogel precursor (polymer solution) to the metal salt to form hydrogel followed by the addition of organic ligands to form in situ growth of the MOF/hydrogel composite (Figure 2A(b)) [20,23,28]. Various MOF-based hydrogels have been prepared using in situ growth methods displayed in Figure 2A [29,30]. The in situ growth method is advantageous in terms of its flexibility, uniform size distribution, high porosity, excellent cytocompatibility, and large adsorption efficiency even without destroying MOF porosity [20,23,27].

#### 3.1.2. One-Step Growth Method

The one-step method involves the simultaneous formation of MOF and hydrogel by adding hydrogel precursor, metal precursor, and organic ligands. The MOF-based hydrogel starts the crosslinking process that occurs in the presence of an organic linker (Figure 2B) [20,27,28]. However, this method is suitable only for MOFs that can be prepared in an aqueous solution at room temperature, not for the MOFs that are usually synthesized by the solvothermal method using organic solvent and high temperature [20]. For instance, zeolitic imidazolate framework (ZIFs) MOFs can be prepared rapidly in aqueous solution at room temperature. Therefore, ZIF-67 [31] and ZIF-8 [32] are commonly used for the preparation of MOF-based hydrogel using the one-step growth method.

#### 3.1.3. Direct Mixing Method

The direct mixing method involves directly adding pre-synthesized MOF to the polymeric hydrogel matrix followed by crosslinking between them through non-covalent and covalent bonds (Figure 2C) [20]. The MOFs can be added to the hydrogel matrix via various methods such as the direct casting of MOFs onto the hydrogel matrix [33], adding MOFs to the prepolymer solution before polymerization [34], and direct mixing into the hydrogel [35]. Moreover, the MOF-based hydrogel can be synthesized by adding a photo-initiator to instigate free radical polymerization in the presence of ultraviolet (UV) or visible light [36]. The direct mixing method is non-toxic, straightforward, and eco-friendly. This synthesis strategy significantly influences the morphology and stability of the MOF-based hydrogels. For example, the direct mixing method may form heterogeneously shaped MOF-based hydrogel due to the non-uniform distribution of MOF into the hydrogel. In contrast, the in situ growth method yields a homogeneously shaped formulation due to the uniform distribution of MOF [27].

### 3.2. Characterization of MOF-Based Hydrogel

Various characterization techniques have been used to confirm the successful formation of the composite with accurate structures. The characterization technique depends on the synthetic strategy of the MOF-based hydrogel. Commonly used techniques are scanning electron microscope/field emission electron microscope (SEM/FESEM) and transmission electron microscopy (TEM) to determine the size, structure, and surface morphology, X-ray diffraction (XRD) to determine the structural integrity, Brunauer–Emmett–Teller (BET) theory to determine the pore size, surface area, and pore volume, Fourier transform infrared spectroscopy (FTIR) to identify the interaction between MOF and polymeric hydrogel, and thermogravimetric analysis (TGA) for thermal stability of the formulations [24,37,38,39]. These techniques are mainly applied to different MOFs but can be extended to MOF-based hydrogels. For example, SEM/TEM is commonly used to visualize how the MOFs embed onto the surface of a polymer matrix. Moreover, energy-dispersive X-ray spectroscopy (EDX) is used to find the elemental distribution within the sample [40]. In a study, FESEM and EDX were used for determining the surface morphology, XRD for crystallinity, and FTIR for chemical group identification of MOF-based hydrogel [41].

Further, the selection of characterization techniques depends on the specific application of the formulation. For instance, cyclic voltammetry is used when the MOF-based hydrogels are utilized as supercapacitors, whereas mechanical and absorptivity properties are characterized for the removal of toxic contaminants application. For biomedical applications, drug loading/in vitro drug-release studies and biocompatibility studies are carried out using spectroscopy studies and cell viability assays, respectively [24]. Additionally, hydrogels are evaluated for rheology by the continuous flow sweep test and oscillatory frequency sweep curve, for degree of swelling by the water absorption test and swelling test, and for mechanical properties by tensile and compressive strength testing [42].

## 4. Recent Advances in MOF-Based Hydrogel for Cancer Management

Recently, extensive research has been carried out to investigate the anticancer potential of MOF-based hydrogels in contrast to MOFs due to their magnified porosity, augmented surface area, customized functionality, biodegradability, and biocompatibility. Moreover, these gels possess the properties of both MOFs and hydrogels [24]. Further, MOF-based hydrogels can also act as multimodal delivery platforms for cancer management using either stimuli-responsive drug delivery, co-delivery of chemotherapeutics, CDT, SDT, ST, and phototherapy, or combination therapy (Table 2). This section emphasizes the applications of different MOF-based hydrogels to treat various cancers.

### 4.1. Breast Cancer

According to the American Cancer Society (ACS), breast cancer is the most common type of cancer in the United States (U.S.) with ~313,510 new cases and ~42,250 deaths. The treatment of breast cancer is dependent on its type, stage, and metastasis and it is usually managed by surgery, radiotherapy, hormone therapy, chemotherapy, and targeted therapies [2]. MOF-based hydrogels are effectively used to manage cancer owing to their distinct properties. In a study, the anticancer property of MOF-based hydrogel was investigated using titanium-based MOF (Ti-MOF) cross-linked with chitosan and oxidized pectin polymer hydrogel matrix. The results revealed that the developed hybrid hydrogel was more cytotoxic against MCF-7 cells (IC_50_: 111 µg/mL) due to the presence of Ti as a bioactive metal and the high surface area and pore volume of Ti-MOF [43]. Moreover, MOF-based hydrogels can also act as multimodal delivery platforms for breast cancer management using stimuli-responsive drug delivery that can localize the drug to the specific target without any toxicity to normal cells. For example, stimuli-sensitive MOF-based hydrogel (MOF-coated with nucleic acid-based polyacrylamide hydrogel) was developed for the controlled release of doxorubicin (DOX). The hydrogel was cross-linked by nucleic acid duplexes that comprise anti-ATP aptamer. It has been observed that the hydrogel was dissociated in the presence of ATP, which is overexpressed in cancer cells, leading to the release of DOX. Moreover, the results revealed that MOF-based hydrogel exhibited higher drug loading with a significant reduction in background drug leakage from the MOF-based hydrogel in contrast to duplex DNA-functionalized MOFs. These unique properties of the stimuli-responsive MOFs/hydrogel carriers lead to improved cytotoxicity toward MDA-MB-231 breast cancer cells as compared to the duplex DNA-capped drug-loaded NMOFs. Further, the developed hydrogel was more selective and cytotoxic against the cancer cells due to the hybridization of the AS1411 aptamer that selectively binds to nucleolin receptors present in cancer cells. Additionally, the drug can be released from the MOF-based hydrogel using other stimuli such as light, metal ions, pH, and microRNA biomarkers [44]. In another study, MOF-based hydrogel was developed by in situ growth synthesis of MOF inside the polymer matrix of carboxymethyl cellulose and graphene quantum dots (CMC/GQDs). The developed hybrid hydrogel was biocompatible and pH-responsive, and it exhibited the augmented loading of DOX (88.90%) in contrast to MOF (MIL-53). Moreover, the hydrogel was effective against breast cancer cells due to nuclei disruption, chromatin condensation, and cell cycle arrest [45]. Recently, a novel HKUST-1-based hybrid hydrogel loaded with 5-fluorouracil (5-FU) was developed for the effective management of breast cancer. The hybrid hydrogel was prepared by the direct mixing of MOF with the polymeric solution of polyvinyl alcohol (PVA) followed by the addition of borax. Small-angle neutron scattering data confirmed that the hydrogel was formed by the cross-linking of PVA with CuBTC using borax. The developed hydrogel exhibited augmented drug loading owing to the large surface area of MOF. Moreover, the developed hydrogel was self-healable, injectable, stretchable, and thermal stable with magnified mechanical properties, and it exhibited pH-responsive drug delivery enabling the specific targeting of HeLa and MCF-7 cancer cells. Further, the developed hybrid hydrogel exhibited successful transportation of the drug followed by the precise release of the drug with magnified cytotoxicity to the cancer cells due to the synergistic interaction of CuBTC, PVA, and borax [46]. In another recent study, a multifunctional MOF-based hydrogel (MIL-100-based hydrogel loaded with 5-FU) was developed for breast cancer management. The hybrid composite was developed by the direct mixing of MIL-100 into the solution of biopolymer (carbomer 940 and xanthan gum) and further characterized using SEM, XRD, FTIR, and strain sweep tests. The results revealed that the developed hybrid hydrogel was self-healable, injectable, and excellent adhesive, and it exhibited excellent swelling and mechanical stability due to the distinct properties of biopolymer. Owing to the unique properties of MIL-100, the developed hybrid hydrogel exhibited magnified drug loading and released the drug in a controlled manner in response to pH variation. Further, the cytotoxicity and results revealed the hybrid hydrogel was biocompatible and exhibited magnified cytotoxicity against MCF-7 cancer cells. In conclusion, the collaborative properties of MOFs and biopolymers can overcome the limitations of individual components and can be effectively used for the management of breast cancer even with minimal toxicity [47]. Interestingly, a pH-responsive MOF-based hydrogel was developed by the cross-linking of magnetic layered double hydroxides functionalized by Cu-based MOF (LDH-Fe_3_O_4_/CuMOF) with the polymer matrix of chitosan followed by the functionalization with к-carrageenan. The developed hydrogel was biocompatible and hemo-compatible, and it exhibited enhanced drug loading (9.6%) with 96.1% encapsulation efficiency along with a controlled release of the drug in a pH-dependent manner due to the presence of LDH and chitosan/к-carrageenan, respectively. Moreover, the hydrogel exhibited enhanced antioxidant activity (71.81%) and cytotoxicity (IC_50_: 52 µg/mL) against MCF-7 cells due to the targeted delivery of DOX suggesting the potential of the MOF-based hydrogel for cancer treatment [48].

Chemotherapy is a traditional way to manage various cancers but suffers from various limitations, especially toxicity and non-specific targeting. Therefore, other novel strategies such as PTT, PDT, CDT, SDT, and ST have been adopted in conjunction with chemotherapy. PTT and PDT rely on the utilization of near-infrared (NIR) light with specific penetration of tumor cells/tissues under light irradiation. PTT transforms the light energy into thermal energy via photothermal agents (PTA), causing irreversible destruction of cancer cells, whereas PDT utilizes photosensitizers (PS) that are induced by specific wavelength light sources, leading to the production of reactive oxygen species (ROS) and resulting in the apoptosis of cancer cells [1]. CDT relies on Fenton or Fenton-like reactions to catalyze tumor microenvironment H_2_O_2_ to toxic OH radicals using divalent metal ions enabling the magnified apoptosis of cancer cells. SDT relies on specific sonosensitizers that are activated by low-intensity ultrasound waves and cause the apoptosis of tumor cells via different mechanisms such as thermal effects, cavitation effects, piezoelectric effects, and sonochemical reactions. ST suppresses tumor growth by hampering the uptake of essential nutrients (glucose and oxygen) and blockage of blood flow [2]. The MOF-based hydrogel can also act as a multifunctional carrier and can also be used for precise and synergistic cancer therapy either using a single strategy or a combination of different strategies. Regarding this, Li et al. developed MnO_2_-functionalized MOF-based hydrogel for mitigating tumor hypoxia with magnified PDT. The hydrogel was developed by the incorporation of MnO_2_-functionalized porphyrin-based MOF into the polymer matrix of chitosan. The results revealed that the developed hydrogel was stable inside the tumor area and exhibited magnified anticancer activity by mitigating tumor hypoxia with enhanced PDT due to the presence of MnO_2_ [49]. Interestingly Kaur et el. developed a curcumin-loaded plasmonic hydrogel for localized chemotherapy and PTT. The hydrogel was based on carboxymethyl chitosan hydrogel cross-linked with ZIF-8 and Mg^2+^. The developed hydrogel was biocompatible, biodegradable, self-healable, and injectable, and it exhibited viscoelastic properties due to the supramolecular interaction between carboxymethyl chitosan and ZIF-8. Moreover, the hybrid hydrogel exhibited enhanced in vitro and in vivo anticancer effects due to the synergistic PTT and chemotherapeutic effect of curcumin [50]. In another study, ZIF-8-based hydrogel was developed for the synergistic PTT-PDT-chemotherapy of breast cancer. The nanocomposite hydrogel was developed with the incorporation of pH-responsive DOX-loaded zeolitic imidazolate frameworks NPs (DOX@ZIF-8 NPs) along with copper sulfide (CuS) NPs into a polymer matrix of CMC and aldehyde-modified methylcellulose (MC-CHO) (Figure 3). The developed hydrogel was characterized and the results revealed that the hydrogel was porous, self-healable, and injectable with excellent swelling property. Moreover, the hydrogel was biocompatible and exhibited PTT and PDT due to the presence of CuS NPs. Moreover, the DOX@ZIF-8 NPs exhibited a controlled release of DOX (in vitro and in vivo) due to the degradation of MOF under an acidic environment. Further, in vitro and in vivo results revealed that the developed hydrogel was cytotoxic against cancer cells and had enhanced anticancer activity due to synergistic PTT-PDT-chemotherapy. In conclusion, the developed MOF-based hydrogel was a multifunctional carrier. It can be safely administered in vivo for slow and controlled release of the drug along with PTT and PDT [51].

Despite its outstanding biocompatibility, pH responsiveness, large surface area, thermal stability at 500 °C, and controlled release at a pH of 7.4, ZIF-8 suffers from certain limitations like disintegration at a pH of 5.4–6 and aggregation in aqueous media. Therefore, ZIF-8 was dispersed into the hydrogel to overcome such limitations. Regarding this, ZIF-based hydrogel was prepared by the direct mixing of zinc-based MOF into CMC and gelatin polymer hydrogel matrix to overcome the initial burst release of quercetin. It has been observed that the developed hydrogel was stable due to its small size (250 nm) and favorable zeta potential (−40.1 mV). Moreover, the hybrid system was pH-responsive and exhibited improved drug release (up to 96 h) without any initial burst effect in the acidic environment of the tumor. Further, the developed formulation was significantly cytotoxic against MCF-7 cells due to the double network of the CMC/gelatin polymer matrix, which can hamper the function of p-glycoprotein efflux pumps [52].

Recently, Zhang et al. developed a MOF-based hydrogel for the management of breast cancer using CDT-PDT-PTT. The hybrid hydrogel was developed by the incorporation of polydopamine functionalized Cu-doped MOF (PCN-224(Cu)@ PDA also known as PCP) into the matrix of CMC and oxidized sodium alginate (OSA). The developed hybrid hydrogel was injectable, self-healable, and biocompatible, and it exhibited enhanced cytotoxicity against 4T1 cells due to the synergistic CDT-PDT-PTT. It has been observed that under NIR irradiation (660 and 808 nm), PCP exhibited PDT and PTT due to the porphyrin core and PDA coating of MOF, respectively. Meanwhile, the CDT effect was due to the conversion of Cu^2+^ into Cu^+^ resulting in the production of free hydroxyl radicals. Moreover, the hydrogel acts as a loading carrier for PCP that can be easily injected into the tumor cells to liberate PCP, followed by the endocytosis to tumor cells, achieving synergistic therapy against cancer (Figure 4) [53].

In another recent study, a smart hybrid hydrogel comprising multifunctional MOF and DOX was developed for breast cancer management. The hybrid hydrogel was developed by the direct mixing of DOX and Au-NPs-decorated Ti/Fe bimetallic MOF tetragonal nanosheets (Au/TF-MOF TNS) into the polymer matrix of hydroxypropyl chitosan (HPCS) and dopamine-grafted sodium alginate (SADA). The developed hydrogel was injectable, self-healable, and pH-responsive, and it exhibited enhanced in vitro and in vivo anticancer activity due to the synergistic SDT-CDT-ST-chemotherapy. The injectability, responses, and self-healability were due to the biopolymers, whereas Au/TF-MOF TNS acts as nano enzymes generating hydroxyl radicals (CDT), oxygen radicals (SDT), and depleted glucose levels (ST). Moreover, the Au/TF-MOF TNS behave as sonosensitizers, transforming the surrounding oxygen into singlet oxygen upon the irradiation of ultrasound to attain SDT (Figure 5) [54].

Interestingly, Zhuang et al. developed a MOF-based hydrogel for magnified breast cancer ferrotherapy via multiplex magnifying redox imbalances. The hybrid hydrogel was developed by the incorporation of paclitaxel/chlorin e6-loaded iron-based MOF (Fe-MOF) into the polymer matrix of calcium alginate. It has been observed that after injection, the alginate can interact with the calcium ions of the cancer site to produce a hydrogel patch, which encourages the confinement of MOF and chemotherapeutics. Moreover, the MOF can magnify tumor redox imbalance via the Fenton reaction and glutathione oxidation owing to its peroxidase/glutathione oxidase mimicking properties. Further, the therapeutics were released upon the ultrasound irradiation and stimulated Ce6 to generate singlet oxygen. Concurrently, a low concentration of paclitaxel can increase lactate generation, resulting in tumoral pH reduction via the downregulation of the glutaminolysis-related gene facilitating the Fenton reaction. Additionally, the downregulation of the glutaminolysis-related gene by paclitaxel further inhibited glutathione synthesis and glutathione peroxidase 4 activity. In conclusion, the hybrid hydrogel exhibited ferrotherapy against breast cancer by magnifying oxidative stress [55].

### 4.2. Colorectal Cancer

According to the ACS, colorectal cancer is the third most common type of cancer excluding skin cancer in the United States (U.S.), with ~106,590 and 46,220 new cases of colon and rectal cancer, respectively [67]. It is usually treated with local (surgery and radiation) and systemic (chemo, targeted, and immunotherapy) treatments. The selection of treatment depends upon the cancer stage and treatments may be used after one another or combined. Chemotherapy is the most preferential strategy for the treatment of colorectal cancer but suffers from various limitations and various drug delivery systems have been developed to overcome the limitations. Anticancer drugs are usually administered either alone or in combination form through an intravenous route, which causes systemic toxicity [2]. Oral delivery is advantageous but suffers from the first-pass metabolism of drugs. MOF-based hydrogels are the most promising carrier for the oral delivery of drugs owing to their distinct properties and stimuli-responsive nature. In a study, MOF-based hydrogel was developed for the effective oral delivery of 5-FU for colon cancer. The hybrid hydrogel was developed by the direct mixing of 5-FU loaded MOF-5 with CMC solution (CMC/5-FU@MOF-5). The results revealed that the CMC/5-FU@MOF-5 released the drug in a controlled manner inside the colon due to the protective and pH-responsive nature of CMC. Moreover, CMC facilitated the transportation of the 5-FU@MOF-5 across the gastrointestinal tract (GIT). Further, the developed hydrogel was significantly effective against HeLa cells [56]. Recently, another MOF-based hydrogel was developed for the effective oral delivery of 5-FU with magnified bioavailability and minimal adverse effects. The hydrogel was developed by the incorporation of mannose (M)-functionalized MOF loaded with 5-FU into the polymer matrix of hyaluronic acid (HA). It has been observed that the developed hydrogel exhibited the pH-responsive release of 5-FU and exhibited magnified cytotoxicity against HT29 cancer cells due to the presence of specific receptors (M and HA), which stimulated targeted drug delivery [57]. In another recent study, M-functionalized MOF-based hydrogel was used to effectively deliver methotrexate against colon cancer. The hybrid was developed by the coating of pectin hydrogel on the surface of M-functionalized MIL-83. The developed hydrogel exhibited the pH-responsive release of the drug and protected the drug from the acidic environment of the stomach due to the protective coating of pectin. Moreover, the hybrid hydrogel exhibited magnified cytotoxicity against HT29 cancer cells due to the presence of M and HA, which stimulated targeted drug delivery [58]. Interestingly, MOF-based hydrogel was developed for dual-functional biotherapy against colon cancer. The hydrogel was based on the incorporation of ZIF-8 loaded with liquid metal NPs into a polymer matrix of alginate. Initially, metal NPs were functionalized with amphiphilic surfactant, allowing the on-demand engineering of the interparticle assembly between MOF and liquid metals, resulting in a supraparticle-like composite material with synergistic multifunctionality and multilevel surface structure, followed by the incorporation in alginate hydrogel. The developed hybrid hydrogel was biodegradable and injectable, and it exhibited a sol–gel transformation with the controlled release of Zn ions leading to the PTT effect, suggesting the anticancer and antibacterial activity of the formulation [59].

Localized delivery can also reduce the side effects because of the direct release of drugs to the target site in a non-invasive manner. The MOF-based hydrogel can be administered locally in the target area followed by the release of the anticancer drug in a sustained manner either through stimuli response or diffusion. Moreover, these hybrid hydrogels minimize the adverse effects due to the bypass of the systemic distribution of drugs along with the improved solubility of drugs owing to the amphiphilic nature of MOFs. Zeng et al. developed an injectable and self-healable MOF-based hydrogel with structurally dynamic properties for the localized delivery of DOX. The hydrogel was developed by the coordination interaction of MOF@DOX with the polymeric matrix of bisphosphonate functionalized HA (Figure 6). The developed hydrogel was biocompatible and ATP- and pH-responsive, and it exhibited enhanced cytotoxicity against CT-26 cells. Moreover, in vivo results revealed that the developed hydrogel significantly suppressed tumor growth with minimal adverse effects in contrast to MOF@DOX due to the localized and sustained delivery of DOX [60].

### 4.3. Miscellaneous

Recently MOF-based hydrogels have been explored to treat various cancers like liver cancer [61,62], oral cancer [35,63], melanoma [64,65], and glioma [66]. Liver cancer is the third main cause of cancer-related deaths, with poor treatment and high incidence [68]. Recently, MOF-based hydrogel has been explored to manage liver cancer due to the unique properties of MOFs and hydrogels, making them multifunctional nanocarriers. Moreover, these smart hydrogels can directly deliver the drug to the target area, enabling a magnified therapeutic efficacy of anticancer drugs. Liu et al. developed a MOF-based hydrogel to effectively deliver the hydrophobic drug curcumin for liver cancer. The hydrogel was based on the in situ growth of ZIF-8 in the polydopamine-functionalized cellulose nanofibril polymer matrix. The result revealed that the developed hydrogel was mechanically strong and released the drug in a controlled manner, enabling the augmented anticancer activity against HepG2 cells [61]. Zhang et al. developed an injectable MOF-based hydrogel for three-in-one anticancer therapy. The hydrogel was developed by incorporating Cu(I)-isopolymolybdate-based MOF (CCUT-1) co-loaded with 5-FU, DOX, and celastrol into the polymer matrix of chitosan-alginate. The results revealed that the developed composite escaped from the antioxidant defense due to the reversible redox nature of Cu(I) ions enabling sustained delivery of the drugs. Moreover, the developed hybrid hydrogel exhibited magnificent drug loading capacity along with enhanced antitumor activity due to the synergistic chemotherapy/CDT/PDT [62].

Recently ACS estimated approximately 58,450 new patients are suffering from oral cancer and ~12,230 will die from oral cancer. Various approaches have been used to treat oral cancer but cause adverse effects; therefore, localized drug delivery is preferred owing to the sustained release of drugs at the target site eliminating adverse effects. MOF-based hydrogels are promising alternatives for the local delivery of drugs owing to the combined properties of MOFs and hydrogel. For example, Tan et al. developed a MOF-based thermosensitive hydrogel for the localized delivery of DOX and celecoxib for oral cancer. The hybrid hydrogel was developed by the integration of IRMOF-3 co-loaded with celecoxib and DOX into the triblock polymer matrix of PLGA-PEG-PLGA. The developed gel was biodegradable and biocompatible, and it can be easily implanted along with the transition of sol into gel owing to the polymer matrix nature. Moreover, the hybrid hydrogel exhibited magnified drug loading and the pH-responsive release of both drugs in a sustained manner, enabling synergistic anticancer activity against oral cancer cells (Figure 7). Celecoxib expediated the anticancer effect of DOX through the reduction of the VEGF level in cancer, resulting in the obstruction of blood vessels growth. Further, the hybrid hydrogel exhibited magnificent in vivo antitumor efficacy including tumor angiogenesis regulation and apoptosis induction due to the synergistic effect of anticancer drugs. In conclusion, the hydrogel steadily delivers both drugs without any burst release and distribution to the periphery [35].

Recently, Yu et al. developed MOF-based hydrogel for the management of oral cancer. The hydrogel was developed by the incorporation of Cd(II)-based MOF loaded with paclitaxel into the polymeric matrix of HA and carboxymethyl chitosan. The developed hydrogel was evaluated for its therapeutic effect on the expression of Ki-67 (cancer cell proliferation marker gene) using an oral cancer cell line (CAL27). It was observed that the polymer matrix augmented the biocompatibility and stability of the MOF. Moreover, the developed hydrogel exhibited enhanced anticancer efficacy via the downregulation of Ki-67 [63].

MOF-based hydrogel has also been used to treat melanoma by regulating melanoma prognostic markers (VEGFa and S100A6). Gao et al. developed an MOF-based hydrogel by the direct mixing of dacarbazine-loaded Cu-based MOF into the polymeric matrix of HA and carboxymethyl cellulose. The developed hydrogel was porous and exhibited a significant reduction in the viability of melanoma cells. Moreover, the hydrogel inhibited the invasiveness of cancer cells via the suppression of VEGFa and S100A6 [64]. Recently, melanoma was treated by magnified anti-angiogenic immunotherapy using ZIF-8- based hydrogel co-loaded with combretastatin A4 and immunostimulatory adjuvant (poly(I: C)). The hydrogel was developed by the direct mixing of MOF into the polymeric solution of hydrogel. The results revealed that ZIF-8 enables the enhanced loading of (CA4) along with the improved stability of immunostimulatory adjuvant (poly(I: C)) against RNAase. Moreover, the developed hybrid hydrogel was injectable and biocompatible, and it exhibited excellent rheological properties along with prolonged retention of CA4. Further, in vivo, results revealed that the developed formulation exhibited enhanced antitumor activity including angiogenesis suppression, blood vessel destruction, and M1 macrophage infiltration promotion even at a reduced dosage [65]. Interestingly, MOF-based hydrogel was developed for the treatment of glioma using SDT and gas therapy. The hydrogel was developed by porphyrin-based MOF cross-linked with a polymer matrix of cysteine functionalized HA using a di sulfide bond, leading to the form of a double network hydrogel structure (Figure 8). The developed hydrogel was injectable and biocompatible, and it exhibited specific targeting to tumor due to the interaction of HA with CD44 receptors present on the tumor cell surface and bypassing the blood–brain barrier to deliver the sonosensitizers directly into the cancer site. Moreover, the hybrid hydrogel reacted with the glutathione (GSH), which is overexpressed in the tumor microenvironment, to liberate MOF and L-cysteine, which magnify SDT via augmenting oxidative stress through GSH depletion. Further, MOF generated ROS and L-cysteine under ultrasound irradiation to liberate H_2_S for gas therapy. In conclusion, the developed hydrogel was effective for the management of glioma along with minimal adverse effects associated with surgery [66].

## 5. Challenges and Future Perspectives

Despite their unique properties, MOF-based hydrogels have not met their fate in clinical applications, due to various limitations. These limitations should be considered when developing MOF-based hydrogels for biomedical applications.

Presently, the direct mixing method is commonly used due to its multiple advantages, but the pre-synthesized MOFs are usually brittle and available in powder form, which can aggregate and poorly interact with the hydrogel, significantly affecting the therapeutic efficacy of the MOF-based hydrogel [22]. Therefore, metals, ligands, and polymers should be carefully selected along with the optimization of different process parameters (pH, drug concentration, reaction time, and temperature) using computational or statistical approaches that can significantly affect the yield, structure, reproducibility, and stability of MOF-based hydrogel. Further, the in situ characterization techniques should be combined with the structure–performance relationship to find a deeper understanding of how structural variation affects the formulation activity. Additionally, more research is required to develop the MOF-based hydrogel using the in situ growth and one-step methods [69].It is difficult to preserve the activity, structural integrity, and stability of MOFs during the synthesis and storage of hybrid hydrogel due to the aqueous instability of MOFs [24]. Therefore, the aqueous stability of MOFs can be enhanced by the functionalization of MOFs or entrapment in protective coatings [24,70].Most synthesis strategies are lab-scale and utilize toxic chemicals, resulting in poor biocompatibility, which restricts the clinical application of MOF-based hydrogels [22]. Further, removing impurities and activating MOF creates difficulty in large-scale production. Therefore, more research is required to develop a sustainable synthesis strategy for industrial-scale production using an eco-friendly greener approach without compromising the quality of the product at a reasonable cost [69]. Moreover, the synthesis of MOF-based hydrogel should comply with current good manufacturing practices [24].Regarding hydrogels, natural polymers may face further challenges due to their natural heterogeneity and poor mechanical properties, resulting in poor stability and reproducibility. Moreover, the rheological properties of hydrogels may vary after the incorporation or integration of MOF, which can affect the biocompatibility and clinical efficacy of MOF-based hydrogel [24,28]. Therefore, natural polymers with good stability should be adopted to overcome such limitations. Regarding the stability of hydrogels, the natural polymers should be functionalized with the Hofmeister salts to provide a cost-effective natural polymer with magnified excellent processability, modifiable degradation, and augmented mechanical properties [28].MOF-based hydrogels can cause adverse effects and long-term toxicity due to the presence of metal ions in MOF, which restricts the clinical application of these hydrogels. Therefore, researchers should focus on the preparation of biocompatible MOF using different strategies like the selection of endogenous molecules and biocompatible metal ions, e.g., Ca, Fe, Mg, etc. [24,28].It has been observed that most MOF-based hydrogels are only evaluated for short-term in vitro cytotoxicity studies that are not satisfactory for further clinical application and commercialization. Therefore, the researchers should focus on in vivo and long-term studies to understand the actual therapeutic mechanism, biodegradation pathways, accumulation in cells/tissues, and adverse effects on human beings [24].

## 6. Conclusions

Recently, various groups have investigated the anticancer potential of MOF-based hydrogels due to their magnified porosity, augmented surface area, customized functionality, biodegradability, and biocompatibility. These hybrid hydrogels are advantageous over individual MOFs and hydrogels due to their combined properties, which complement each other and overcome the others’ limitations. For instance, hydrogels ensure the uniform distribution of MOF within the polymer matrix bypassing aggregation and can also act as a drug reservoir and control the release of drugs, whereas MOFs can augment the mechanical properties of the hydrogels, enabling their suitability for biomedical application. These hydrogels are usually prepared by the introduction of MOFs into the polymeric hydrogel matrices by the in situ growth, one-step, and direct mixing methods. This review briefly describes the different strategies used for the synthesis and characterization of MOF-based hydrogels. Further, special emphasis has been given to the recent advances in MOF-based hydrogels used to manage different cancers. Various MOF-based hydrogels have been developed for the management of different cancers using diverse MOFs (Ti MOF, Uio-68, MIL-53, MIL-100, HKUST-1, Cu-MOF, PCN-224, ZIF-8, Au/TF-MOF TNS, MOF-5, MIL-88(Fe), CCUT-1, IRMOF-3, and Cu-MOF-cys) and various biopolymers. It has been observed that MOF-based hydrogel can also act as a multifunctional carrier and can also be used for precise and synergistic cancer therapy either using stimuli-responsive drug delivery, the co-delivery of chemotherapeutics, CDT, SDT, ST, and phototherapy, or combination therapy. Further, certain hydrogels are injectable and self-healable, and they can also be used for the oral and localized delivery of chemotherapeutics owing to their distinct properties and stimuli-responsive nature. It can be concluded that MOFs are responsible for the controlled and stimuli-responsive drug release whereas hydrogel provides the biocompatibility and stability of MOFs in biological media. However, these hybrid hydrogels are not clinically used and still at an early stage due to certain limitations like crucial and small-scale synthesis, limited in vivo studies, and limited long-term toxicity studies. Therefore, extensive research is required to develop a sustainable synthesis strategy for industrial-scale production using an eco-friendly green approach without compromising the quality of the product at a reasonable cost. Moreover, these hydrogels should be characterized in a timely manner using XRD and nitrogen isothermal adsorption to assess their stability. More in vivo studies are required to understand the actual therapeutic mechanism, biodegradation pathways, and adverse effects on human beings. We believe that this review will provide a comprehensive holistic view to researchers for developing MOF-based hydrogels with augmented anticancer effects, enabling the effective management of cancer even without adverse effects.

## Figures and Tables

**Figure 1 gels-11-00076-f001:**
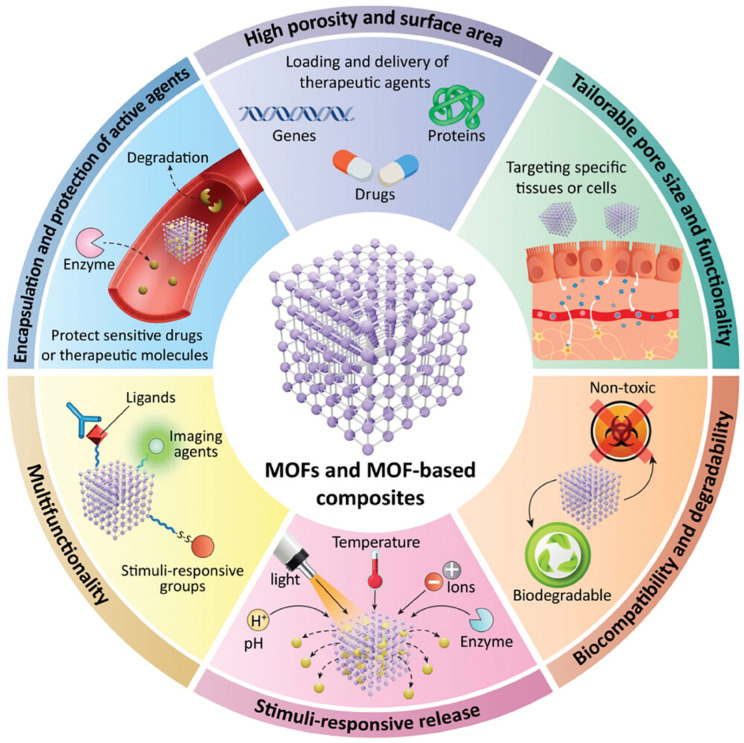
Distinct properties of metal–organic frameworks (MOFs) and MOF-based composites. Adapted from [26], Wiley (2024).

**Figure 2 gels-11-00076-f002:**
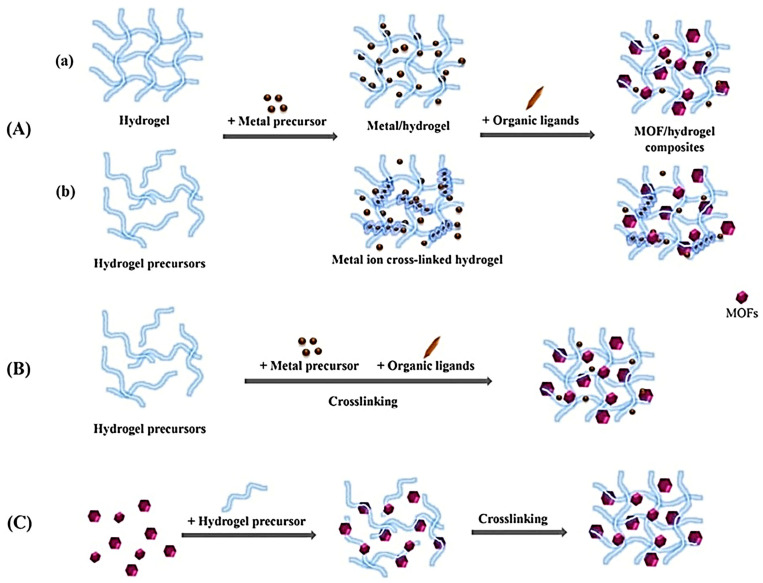
Synthesis of metal–organic framework (MOF)-based hydrogel using (**A**) in situ growth method; (**B**) simultaneous formation of MOF and hydrogel; and (**C**) direct mixing method. Reprinted from [20], with permission from Elsevier.

**Figure 3 gels-11-00076-f003:**
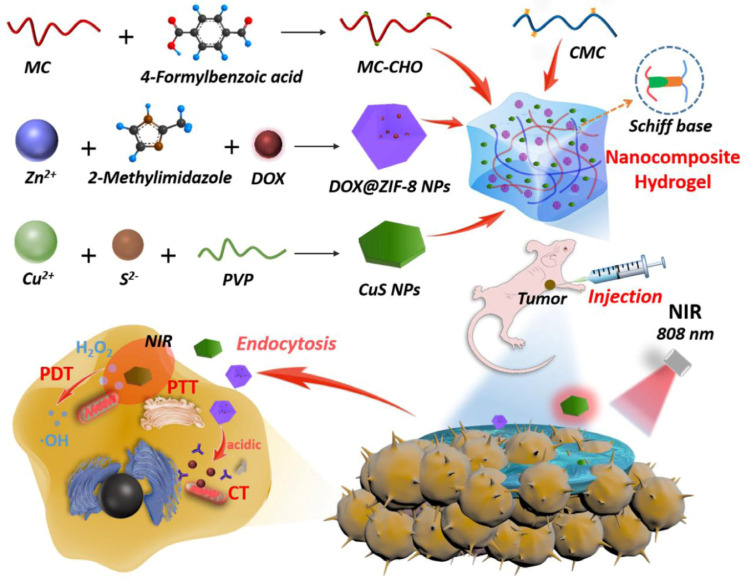
Development of nanocomposite hydrogel for the synergistic PTT-PDT-chemotherapy of breast cancer. Reprinted from [51], with permission from Elsevier.

**Figure 4 gels-11-00076-f004:**
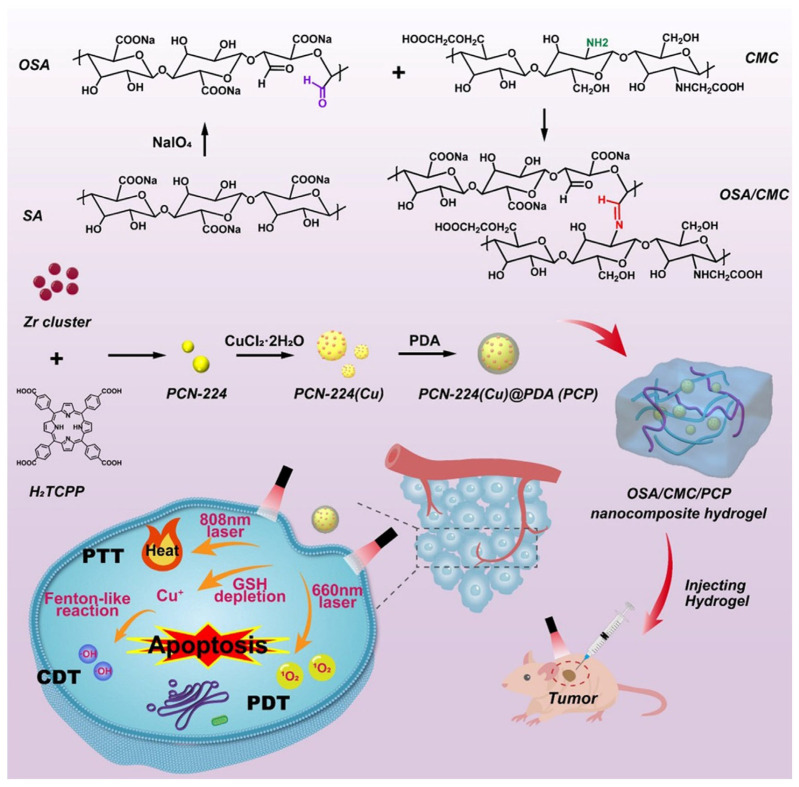
Preparation of MOF-based hydrogel for the synergistic therapy of breast cancer using chemodynamic-photodynamic-photothermal therapy. Reprinted from [53], with permission from Elsevier.

**Figure 5 gels-11-00076-f005:**
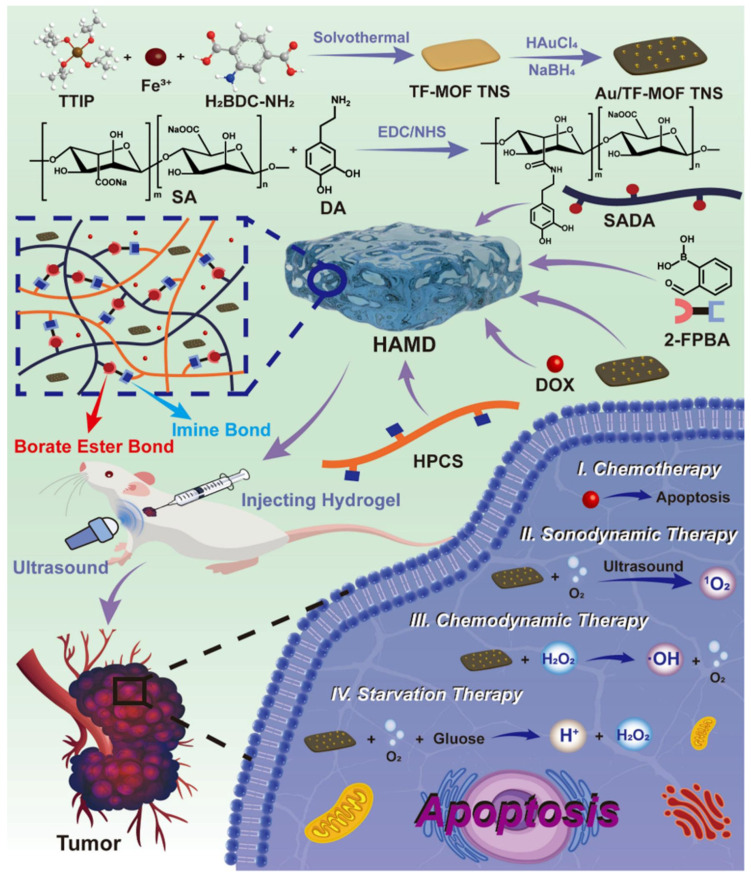
Smart MOF-based hydrogel for the effective management of breast cancer using synergistic SDT-CDT-ST-chemotherapy. Reprinted from [54], with permission from Elsevier.

**Figure 6 gels-11-00076-f006:**
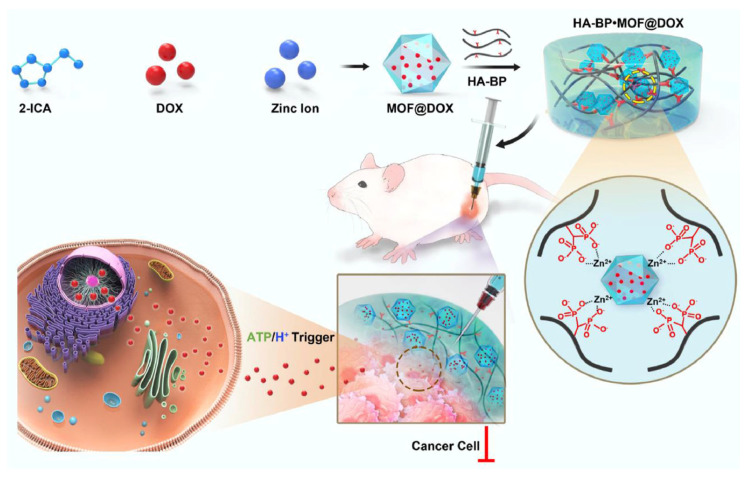
MOF-based hydrogel for the localized delivery of DOX against colon cancer. Reprinted from [60], with permission from Elsevier.

**Figure 7 gels-11-00076-f007:**
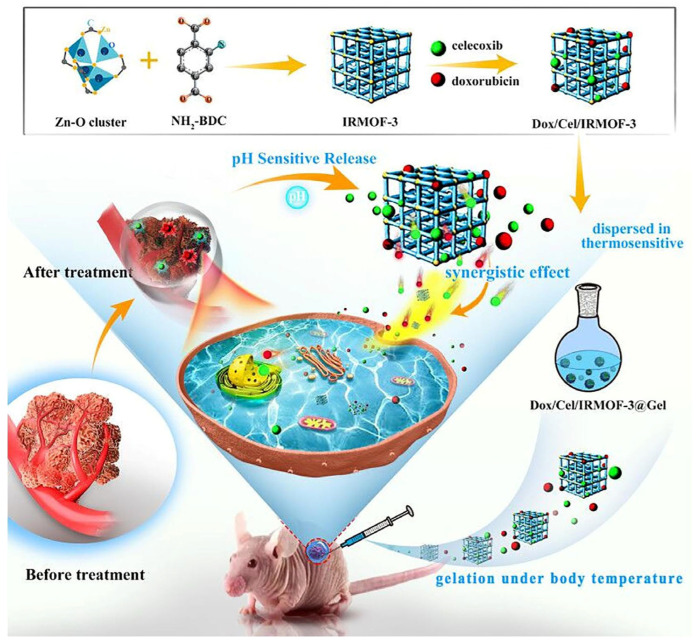
Multifunctional MOF-based thermosensitive hydrogel co-loaded with doxorubicin and celecoxib for synergistic oral anticancer activity. Reprinted from [35], with permission from Elsevier.

**Figure 8 gels-11-00076-f008:**
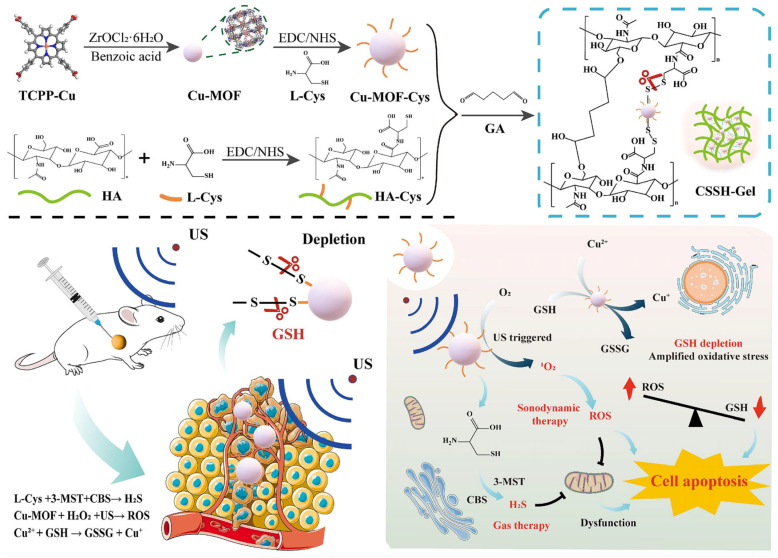
Fabrication and anticancer mechanism of porphyrin-based hydrogel for the management of glioma. Reprinted from [66], with permission from Elsevier.

**Table 1 gels-11-00076-t001:** Summary of different strategies used for the synthesis of MOF-based hydrogels.

Synthesis Strategy	Technique	Advantages	Limitations	Examples MOF/Polymer	Ref
In situ growth	Dispersion of metal precursor into pre-synthesized hydrogel followed by the attachment of metal ions in the pores of the hydrogel via chemical or physical interaction. Then, organic ligands are added, leading to the contact with the metal ions via coordination chemistry resulting in the formation of MOF into the pores of a pre-prepared hydrogel.	Flexible and easy synthesis with a uniform size distribution, high porosity, excellent cytocompatibility, and large adsorption efficiency even without the destruction of MOF porosity.	Restricted in situ synthesis of MOF that can be synthesized under specific conditions. Moreover, cannot precisely control the MOF loading into the hydrogel matrix to obtain the required MOF–hydrogel ratio.	HKUST-1, ZIF-8, MIL-100(Fe), and ZIF-67/alginate and ZIF-8/PVA-sodium alginate	[20,23,27,29,30]
One-step method	The precursors of hydrogels and MOFs are mixed to form simultaneous formation of both followed by cross-linking with each other.	Mild reaction conditions, rapid, and excellent adsorption efficiency.	Suitable only for MOFs that can be prepared in aqueous solution at room temperature.	ZIF-67/alginate and ZIF-8/cellulose	[20,27,28,31,32]
Direct mixing	Pre-synthesized MOF are added to the polymeric hydrogel through various methods like physical mixing and casting followed by the self-cross-linking of MOF with hydrogel matrix via hydrogen bond, metal-coordination bonds, and van der Waals forces.	Easy, mild experimental conditions with precise loading of MOF into the hydrogel matrix. Moreover, MOF structural integrity can be maintained along with its unique properties.	Weak interaction between MOF and hydrogel and heterogenously shaped formulation along with limited interfacial viscosity.	UiO-66-NO_2_/gelatin, ZIF-8/acrylamide, and IRMOF-3/PLGA-PEG-PLGA	[20,23,24,27,28,33,34,35]

Abbreviations: HKUST: Hong Kong University of Science and Technology; IRMOF: isoreticular metal–organic framework; MIL: Materials of Institute Lavoisier; PEG: polyethylene glycol; PLGA: poly(lactic-co-glycolic acid); PVA: polyvinyl alcohol; ZIF: zeolitic imidazolate frameworks.

**Table 2 gels-11-00076-t002:** Summary of recently used MOF-based hydrogels for the management of various cancers.

MOF	Metal Ion/Ligand	Polymeric Hydrogel Matrix	Synthesis of MOF-Based Hydrogel	Drug	Cancer	Anticancer Mechanism	Strength	Ref
Ti-MOF	Titanium nitrate tetrahydrate/2,6 pyridine dicarboxylic acid	Oxidized pectin and chitosan	Direct mixing	-	Breast cancer	Anticancer property of titanium	The developed hydrogel exhibited enhanced cytotoxicity against MCF-7 cells owing to the presence of bioactive metal Ti.	[43]
UiO-68	ZrCl_4_/amino-triphenyl dicarboxylic acid	Polyacrylamide	Direct mixing	DOX		Chemotherapy and stimuli-responsive	The developed formulation was advantageous in terms of higher drug loading, non-specific leakage of drug along with higher cytotoxicity against MD-MBA-231 cells due to the hybridization of AS1411 aptamer that selectively binds to nucleolin receptors present in cancer cells.	[44]
MIL-53	Al(NO_3_)_3_·9H_2_O/Na_2_BDC	CMC/GQD	In situ growth	DOX		Chemotherapy and pH-responsive	The developed hybrid hydrogel was biocompatible and exhibited magnified cytotoxicity against MDA-MB 231 cells due to nuclei disruption, chromatin condensation, and cell cycle arrest.	[45]
HKUST-1	Cu(NO_3_)_2_·3H_2_O/BTC	PVA and borax	Direct mixing	5-FU		Chemotherapy and pH-responsive	The developed hybrid hydrogel successfully transported the drug followed by its precise release, magnifying its cytotoxicity to cancer cells due to the synergistic interaction of CuBTC, PVA, and borax.	[46]
MIL-100	Fe(NO_3_)_3_·9H_2_O/BTC	Xanthan gum and Carbomer 940	Direct mixing	5-FU		Chemotherapy and pH-responsive	The developed hybrid hydrogel was self-healable, injectable, adhesive, biocompatible, and mechanically stable; it exhibited magnified drug loading and released the drug in a controlled manner in response to pH variation along with magnified cytotoxicity against MCF-7 cancer cells. Management of breast cancer even with minimal toxicity.	[47]
Cu-MOF	Cu(NO_3_)_2_·3H_2_O/TPA	Chitosan	Direct mixing	DOX		Chemotherapy and pH-responsive	The developed hydrogel was biocompatible and hemocompatible, and exhibited enhanced antioxidant activity and cytotoxicity against MCF-7 cells due to the presence of LDH and chitosan/к-carrageenan.	[48]
PCN-224	ZrOCl_2_·8H_2_O/TCPP	Chitosan	Direct mixing	-		PDT	The developed hydrogel was stable inside the tumor area and exhibited magnified anticancer activity by mitigating tumor hypoxia with enhanced PDT due to the presence of MnO_2_.	[49]
ZIF-8	Zn(NO_3_)_2_·6H_2_O/2-methylimidazole	Carboxymethyl chitosan	Direct mixing	Curcumin		Chemotherapy and PTT	The developed hydrogel was biocompatible, biodegradable, self-healable, and injectable, and exhibited enhanced in vitro and in vivo anticancer effects due to the synergistic PTT and chemotherapeutic effect of curcumin.	[50]
ZIF-8	Zn(NO_3_)_2_·6H_2_O/2-methylimidazole	MC -CHO and CMC	Direct mixing	DOX		PTT-PDT-chemotherapy, and pH-responsive	The developed nanohybrid exhibited magnificent anticancer activity due to the synergistic effect of controlled release of DOX (in vitro and in vivo) and the photodynamic activity of CuS NPs along with the retention of the self-healing hydrogel.	[51]
ZIF-8	Zn(NO_3_)_2_·6H_2_O/2-methylimidazole	CMC-gelatin	Direct mixing	Quercetin		Chemotherapy and pH-responsive	The developed hybrid system was biocompatible and exhibited pH-responsive drug release and higher cytotoxicity against MCF-7 cells due to the double network of CMC/gelatin polymer matrix which can hamper the function of p-glycoprotein efflux pumps.	[52]
PCN-224 (Cu)	CuCl_2_·2H_2_O/PCN-224	OSA-CMC	Direct mixing	-		CDT-PDT-PTT	The developed hybrid hydrogel was injectable, self-healable, and exhibited PDT and PTT due to the porphyrin core and PDA coating of MOF, respectively. Moreover, the hybrid hydrogel exhibited a CDT effect due to the conversion of Cu^2+^ into Cu ^+^ resulting in the production of free hydroxyl radical.	[53]
Au/TF-MOF TNS	FeCl_3_·6H_2_O/H_2_BDC-NH_2_ and TTIP	SADA-HPCS	Direct mixing	DOX		SDT-CDT-ST-chemotherapy	The developed smart hydrogel was injectable, self-healable, and pH-responsive, and exhibited SDT-CDT-ST-chemotherapy effect against tumor. Moreover, the Au/TF-MOF TNS behave as sonosensitizers, transforming the surrounding oxygen into singlet oxygen upon the irradiation of ultrasound to attain SDT.	[54]
Fe-MOF	FeCl_3_·6H_2_O/H_2_BDC-NH_2_	Calcium-alginate	Direct mixing	Paclitaxel		Ferrotherapy and chemotherapy	The hybrid hydrogel exhibited ferrotherapy against breast cancer via multiplex magnifying redox imbalances.	[55]
MOF-5	Zn(NO_3_)_2_·6H_2_O/BDC	CMC	Direct mixing	5-FU	Colorectal cancer	pH-responsive and chemotherapy	The developed hydrogel was effective against the cancer cells and released the drug in a controlled manner along with the facilitated transportation of 5-FU@MOF-5 owing to the CMC.	[56]
MIL-88(Fe)	Fe(NO_3_)_3_·9H_2_O/TPA	HA	Direct mixing	5-FU		pH-responsive, chemotherapy, and targeted delivery	The developed hydrogel exhibited pH-responsive release of 5-FU and exhibited magnified cytotoxicity against HT29 cancer cells due to the presence of specific receptors (M and HA), which stimulated targeted drug delivery.	[57]
MIL-88(Fe)	Fe(NO_3_)_3_·9H_2_O/TPA	Pectin	Direct mixing	Methotrexate		pH-responsive, chemotherapy, and targeted delivery	The developed hydrogel was cytocompatible due to the biocompatible pectin. Moreover, it exhibited a pH-responsive release of methotrexate along with magnified cytotoxicity against HT29 cancer cells due to the presence of M receptors which stimulated targeted drug delivery.	[58]
ZIF-8	Zn(NO_3_)_2_·6H_2_O/2-methylimidazole	Alginate	Direct mixing	EGaIn LMNPs		Chemotherapy	The developed hybrid hydrogel was biodegradable, injectable, and exhibited the PTT effect and controlled release of Zn ions suggesting the anticancer and antibacterial activity of formulation.	[59]
MOF	Zinc acetate/ICA	HA-BP	Coordination cross-linking	DOX		ATP-, pH-responsive, and chemotherapy	The developed hydrogel was stimuli-responsive and exhibited localized delivery of anticancer drugs due to the hydrogel network.	[60]
ZIF-8	Zn(NO_3_)_2_·6H_2_O/2-methylimidazole	polydopamine-functionalized cellulose nanofibril	In situ growth	Curcumin	Liver cancer	Chemotherapy	The developed composite hydrogel exhibited enhanced anticancer activity against HepG2 cells due to the controlled release of curcumin.	[61]
CCUT-1	Cu(I)-isopolymolybdate/TBA	Chitosan-alginate	Direct mixing	5-FU, DOX, and celastrol		Chemotherapy/CDT/PDT	The developed hybrid hydrogel exhibited magnificent drug loading capacity and released the drug in a sustained manner leading to enhanced antitumor activity.	[62]
IRMOF-3	Zn(NO_3_)_2_·6H_2_O/NH_2_-BDC	PLGA-PEG-PLGA	Direct mixing	DOX and celecoxib	Oral cancer	Chemotherapy and pH-responsive	The developed implantable gel was thermosensitive, biodegradable, and biocompatible, and exhibited pH-responsive release of the drugs in a sustained manner enabling synergistic anticancer activity against SCC9 and KB cells.	[35]
Cd(II)-based MOFs	Cd(NO_3_)_2_·6H_2_O/H_3_L_1_ and CdCl_2_·2H_2_O/H_3_L_2_	HA/carboxymethyl chitosan	Direct mixing	Paclitaxel		Chemotherapy	The developed hydrogel exhibited enhanced anticancer efficacy via the downregulation of Ki-67 against CAL27 cells.	[63]
Cu-MOF	Cu(NO_3_)_2_·3H_2_O/H_4_btca	HA/carboxymethyl chitosan	Direct mixing	Dacarbazine	Melanoma	Chemotherapy	The developed hydrogel exhibited a significant reduction in the viability of melanoma cells and inhibited the invasiveness of cancer cells via the suppression of VEGFa and S100A6.	[64]
ZIF-8	Zn(NO_3_)_2_·6H_2_O/2-methylimidazole	HA	Direct mixing	CA4 and poly(I: C)		Immunotherapy	The developed hybrid hydrogel was injectable, biocompatible, and exhibited prolonged retention of CA4 enabling enhanced antitumor activity even at a reduced dosage.	[65]
Cu-MOF-cys	ZrOCl_2_⋅8H_2_O/TCPP-Cu	HA-cys	Direct mixing	-	Glioma	SDT and gas therapy	The developed hydrogel was injectable, biocompatible, and exhibited specific targeting to tumors due to the interaction of HA with CD44 receptors, and it released the sonosensitizers, enabling magnified gas therapy and SDT.	[66]

Abbreviations: 5-FU: 5-fluorouracil; Au: Gold; BDC: 1,4-benzene dicarboxylate; BTC: 1,3,5 benzene tricarboxylic acid; CA4: combretastatin A4; CCUT-1: Cu(I)-isopolymolybdate-based MOF; CDT: chemodynamic therapy; CMC: carboxymethyl cellulose; CuS: copper sulfide; DOX: doxorubicin; EGaIn: eutectic gallium indium; Fe(NO_3_)_3_⋅9H_2_O: Iron (III) nitrate nonahydrate; FeCl_3_⋅6H_2_O: iron chloride hexahydrate; GQDs: graphene quantum dots; H_2_BDC-NH_2_: 2-aminoterephthalic acid; H_3_L_1_: 4,4′,4″-((benzene-1,3,5-tricarbonyl) tris(azanediyl)) tribenzoic acid; H_3_L_2_: 4,4′,4″-(benzene-1,3,5-triyltris(azanediyl))tribenzoic acid): H_4_btca: 1,2,4,5 benzenetetracarboxylic acid; HA: hyaluronic acid; HA-BP: bisphosphonate functionalized HA; HKUST-1: Hong Kong University of Science and Technology-1; HPCS: hydroxypropyl chitosan; ICA: Imidazole-2-carboxaldehyde; LMNPs: liquid metal nanoparticles; MC-CHO: aldehyde-modified methylcellulose; MIL-100: MOFs: metal–organic frameworks; NPs: nanoparticles; OSA: oxidized sodium alginate; PDT: photodynamic therapy; PEG: -poly(ethylene glycol); PLGA: poly(D,L-lactide-coglycolide); PTT: photothermal therapy; PVA: polyvinyl alcohol; SADA: grafted sodium alginate; SDT: sonodynamic therapy; ST: starvation therapy; TBA: 1,2,4-triazole; TCPP: Meso-Tetra(4-carboxyphenyl)porphine; TCPP-Cu: [5,10,15,20-Tetrakis(4-carboxyphenyl)porphyrinato]copper(II); Ti: titanium; TNS: tetragonal nanosheets; TPA: terephthalic acid; TTIP: Titanium (IV) isopropoxide; UiO-68: Universitetet i Oslo-68; ZIF: zeolitic imidazolate frameworks; Zn(NO_3_)_2_·6H_2_O: zinc nitrate hexahydrate; ZrCl_4_: zirconium chloride.

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
