# Peer review of "Recent Advances in Metal–Organic Framework-Based Anticancer Hydrogels"

_gels, 2025, doi:10.3390/gels11010076_

Round 1
Reviewer 1 Report
Comments and Suggestions for Authors
Manuscript ID: gels-3416303
Title: Recent advances in metal-organic framework-based anticancer hydrogels
Comments:
Kush and coworkers describe the recent advances in metal organic framework based anticancer hydrogels. The review is interesting, however, following points need to be addressed:
Page 1, line 20, Page 3, line 102, Page 3, lines 113-115: The statements require to be reviewed.
The authors are recommended citing more relevant literature to support any claims made throughout the article. For instance, the authors discussed different strategies for treating cancer or mentioned various types of polymers employed in the preparation of hydrogel matrix without providing proper references.
Page 3, line 130: The authors state that “The in situ growth method involves the direct formation of MOF into the pores of a pre-prepared hydrogel.” However, the respective figure (Part b of Fig 1, A) indicates hydrogel precursors. Could the authors elaborate the statement for better apprehension?
The readers would find it of great interest if the authors could rearrange table 1 by including examples and some information regarding the experimental conditions. In this way, it would be more clear to show the strategies used to synthesize MOF based hydrogel. The data currently displayed in the table needs to be more specific.
In one step method, the authors describe that “suitable only for MOFs that can be prepared in aqueous solution at room temperature”. Could the authors provide examples to elaborate this statement?
Could the authors include more description on the fabrication of MOF based hydrogels in the relevant section.
I would suggest to exclude table 2, as it does not contain any significant information. However, including more specific details about characterization techniques in the respective text would be beneficial for the readers.
The author described the recent advances in MOF based hydrogels in the tabular form (Table 3). However, this data deserves to be discussed in a more straightforward and elaborative way and presented in a proper section.
In table 3, the authors reported mostly examples of MOF based hydrogels synthesized through direct mixing. It would be more helpful for the readers if the authors could provide examples that cover all the synthetic methods. Could the authors provide a comment on the frequent use of direct mixing method?
The MOF based hydrogels mentioned in the article mainly utilize modified MOF systems. Could the authors explain the effect of modification of MOFs on the anticancer mechanism of MOF based hydrogels?
Page 30, lines 340-342: The reference format needs to be reviewed.
Page 34, line 354: I believe that the authors are referring to ‘in situ growth’.
Page 35, lines 494-495: The statement needs to be reviewed.
Page 35, line 497: UiO-68
Page 35-36, lines 508-511 and 512-515: There is a repetition of the statement.
The authors are advised to check the English of the manuscript for grammar errors, punctuation marks, and typos.
Author Response
To Date: 11.01.2025
Editorial Department of Gels
Dear Editor,
I am thankful for a positive response from your office on our manuscript (gels-3416303). Special thanks are due for a speedy review process which has further improved the quality of this work. The authors have addressed all the suggestions and comments. The description of the changes made is given as follows:
Reviewer #1:
Kush and coworkers describe the recent advances in metal organic framework based anticancer hydrogels. The review is interesting, however, following points need to be addressed:
Page 1, line 20, Page 3, line 102, Page 3, lines 113-115: The statements require to be reviewed.
Response: The lines have been thoroughly revised and highlighted in red.
The authors are recommended citing more relevant literature to support any claims made throughout the article. For instance, the authors discussed different strategies for treating cancer or mentioned various types of polymers employed in the preparation of hydrogel matrix without providing proper references.
Response: All the necessary literature has been cited appropriately and highlighted in red. References related to the polymers have been mentioned in section 4 and Table 2.
Page 3, line 130: The authors state that “The in situ growth method involves the direct formation of MOF into the pores of a pre-prepared hydrogel.” However, the respective figure (Part b of Fig 1, A) indicates hydrogel precursors. Could the authors elaborate the statement for better apprehension?
Response: The paragraph has been thoroughly modified and highlighted in red.
The readers would find it of great interest if the authors could rearrange table 1 by including examples and some information regarding the experimental conditions. In this way, it would be more clear to show the strategies used to synthesize MOF based hydrogel. The data currently displayed in the table needs to be more specific.
Response: Table 1 has been modified and the relevant examples have been added. Moreover, we have tried to include some information such as MOF and polymer matrix used for the preparation of the MOF-based hydrogel for individual methods.
In one step method, the authors describe that “suitable only for MOFs that can be prepared in aqueous solution at room temperature”. Could the authors provide examples to elaborate this statement?
Response: This subsection has been thoroughly revised and suitable examples have been added.
Could the authors include more description on the fabrication of MOF based hydrogels in the relevant section.
Response: The section has been thoroughly revised and modified according to the suggestions and highlighted in red.
I would suggest to exclude table 2, as it does not contain any significant information. However, including more specific details about characterization techniques in the respective text would be beneficial for the readers.
Response: Table 2 has been removed and the details about the characterization techniques were included in section 3.2.
The author described the recent advances in MOF based hydrogels in the tabular form (Table 3). However, this data deserves to be discussed in a more straightforward and elaborative way and presented in a proper section.
Response: Table 3 (Now table 2) data has been elaborated in the section 4.
In table 3, the authors reported mostly examples of MOF based hydrogels synthesized through direct mixing. It would be more helpful for the readers if the authors could provide examples that cover all the synthetic methods. Could the authors provide a comment on the frequent use of direct mixing method?
Response: Table 3 (now Table 2) presents the summary of recently used MOF-based hydrogel for managing various cancers. We have covered all the literature related to the same content, we found that the direct mixing method is more prevalent due to its multiple advantages such as easy, mild experimental conditions with precise loading of MOF into the hydrogel matrix, and maintenance of structural integrity of MOF.
The MOF based hydrogels mentioned in the article mainly utilize modified MOF systems. Could the authors explain the effect of modification of MOFs on the anticancer mechanism of MOF based hydrogels?
Response: Surface functionalization of MOFs can magnify the versatility of MOFs in terms of augmenting their targeting efficiency, biocompatibility, and dispersibility. Moreover, surface functionalization can also improve the stability of MOFs and release the drug in a controlled manner. In the case of cancer surface functionalization improves the bioavailability of the drug by magnified cellular uptake followed by successful transportation of MOFs with stimuli-responsive and controlled release properties. The same is added in the text.
Page 30, lines 340-342: The reference format needs to be reviewed.
Response: It has been changed.
Page 34, line 354: I believe that the authors are referring to ‘in situ growth’.
Response: yes, it is in situ growth, same is modified.
Page 35, lines 494-495: The statement needs to be reviewed.
Response: The lines have been modified.
Page 35, line 497: UiO-68
Response: The sentence has been modified and highlighted in red.
Page 35-36, lines 508-511 and 512-515: There is a repetition of the statement.
Response: The lines have been modified.
The authors are advised to check the English of the manuscript for grammar errors, punctuation marks, and typos.
Response: The manuscript has been thoroughly revised.
Thank you very much for your consideration.
Yours Sincerely,
Dr. Preeti Kush
Reviewer 2 Report
Comments and Suggestions for Authors
This review discusses the synthesis, characterization, and applications of MOF-based hydrogels. Based on recent studies on their use in cancer treatment, the authors explore the challenges and future prospects of MOF-based hydrogels. So, it is suitable for publication in the journal "Gels" since it has the interesting topic and results. However, it has the following revised parts. They should be checked prior to the publication. Followings are recommended for the revision.
Major revision
1. In the "Recent Advances in MOF-Based Hydrogel for Cancer Management" section, provide more detailed experimental data for all cases to enhance the reliability of the manuscript.
2. Page 32, Add at least two diverse case references for each type of cancer, including oral cancer and melanoma.
Minor revision
1. Figure 1, Enhance the clarity and resolution to improve its visual quality.
2. In Table 2 and Table 3, The formatting is unclear, such as uneven text spacing and inconsistent alignment. Please correct this to improve readability.
3. Figure 1-7, Adjust the size to be consistent across all figures and ensure they are appropriately aligned with the text for better visual coherence.
4. Page6, line 161/ Page13,14 table3 / Page27, line 284,291,292,294/ Page 31, line 382/ Page33, line 420,432/ Page35, line 463,479,508/ Page 36, line515, Italicize in vitro and in vivo wherever they appear in these sections.
5. Page27, line 280, ‘plansmonic hydrogel’ should be corrected to ‘plasmonic hydrogel.’
6. Page28, line 299, Correct the special character in the temperature value to "500 ºC.
7. Page 30, line 342, check the reference format.
Author Response
To Date: 11.01.2025
Editorial Department of Gels
Dear Editor,
I am thankful for a positive response from your office on our manuscript (gels-3416303). Special thanks are due for a speedy review process which has further improved the quality of this work. The authors have addressed all the suggestions and comments. The description of the changes made is given as follows:
Reviewer #2:
Major revision
1. In the "Recent Advances in MOF-Based Hydrogel for Cancer Management" section, provide more detailed experimental data for all cases to enhance the reliability of the manuscript.
Response: More experimental information has been updated in section 4.
2. Page 32, Add at least two diverse case references for each type of cancer, including oral cancer and melanoma.
Response: Thanks for the suggestion. We searched again through different search engines like Scopus, Google Scholar, PubMed Central, Science Direct, and PubMed using the keywords MOF, hydrogel, and melanoma/oral cancer and found one more article and added the same.
Minor revision
1. Figure 1, Enhance the clarity and resolution to improve its visual quality.
Response: Figure 1 (now Figure 2) have been modified.
2. In Table 2 and Table 3, The formatting is unclear, such as uneven text spacing and inconsistent alignment. Please correct this to improve readability.
Response: Table 2 has been removed as per the suggestion of reviewer 1 and Table 3 (now Table 2) has been modified.
3. Figure 1-7, Adjust the size to be consistent across all figures and ensure they are appropriately aligned with the text for better visual coherence.
Response: Thanks for the suggestion, but the editing of figures was done by the journal according to its format. Moreover, we have tried to do the needful. The high-resolution images have been shared with the journal.
4. Page6, line 161/ Page13,14 table3 / Page27, line 284,291,292,294/ Page 31, line 382/ Page33, line 420,432/ Page35, line 463,479,508/ Page 36, line515, Italicize in vitro and in vivo wherever they appear in these sections.
Response: Corrected
5. Page27, line 280, ‘plansmonic hydrogel’ should be corrected to ‘plasmonic hydrogel.’
Response: It has been corrected.
6. Page28, line 299, Correct the special character in the temperature value to "500 ºC.
Response: Thanks for the correction, changed.
7. Page 30, line 342, check the reference format.
Response: It has been changed.
Thank you very much for your consideration.
Yours Sincerely,
Dr. Preeti Kush
Reviewer 3 Report
Comments and Suggestions for Authors
The review conducted by Kush and Kumar et al. provides a comprehensive overview of metal-organic framework (MOF)-based hydrogels for cancer management. The authors effectively synthesize current research on this promising therapeutic approach, covering synthesis methods, characterization techniques, and recent advances in applying MOF-based hydrogels to various cancer types. Key highlights include detailed explanations of synthesis strategies, thorough discussion of characterization techniques, extensive coverage of applications for different cancers (particularly breast and colorectal cancer), insights into multifunctional approaches combining chemotherapy with photothermal, photodynamic, and other therapies, and a balanced view of current challenges and future perspectives for clinical translation. The authors present complex information in an accessible manner, enhanced by figures and tables, though some images could be improved for clarity. This review contributes significantly to the literature on MOF-based hydrogels for cancer treatment, serving as an important reference for scientists working in this rapidly evolving area of cancer therapeutics.
However, while the review is interesting and innovative, several points need attention from the authors:
Could you provide more details on the search strategy, including specific search terms, date ranges, and inclusion/exclusion criteria for the selected studies?
While the review provides a comprehensive overview of MOF-based hydrogels for cancer treatment, it would benefit from a more critical evaluation of the studies' strengths and limitations.
Some figures, particularly Figures 2-7, appear low-resolution. Please provide higher-quality images to enhance readability.
Discuss any potential biases in the selected studies and how they might impact the interpretation of results.
Expand the discussion on the challenges and opportunities for translating MOF-based hydrogels from preclinical studies to clinical applications.
Include a section that directly compares the efficacy and safety of MOF-based hydrogels with other existing cancer treatments or drug delivery systems.
Given the rapidly evolving nature of this field, please make sure that the review includes the most recent relevant studies (up to 2024).
Based on the gaps identified in the current literature, offer more specific recommendations for future research directions.
Consider adding a summary table that compares key features (e.g., synthesis method, cancer type, therapeutic mechanism) of the different MOF-based hydrogels discussed in the review.
The introduction could be enhanced by incorporating more detailed information. For example, the author can add more details using the following paper and referencing it:
https://www.sciencedirect.com/science/article/abs/pii/S1742706123006025
Sincerely,
Author Response
To Date: 11.01.2025
Editorial Department of Gels
Dear Editor,
I am thankful for a positive response from your office on our manuscript (gels-3416303). Special thanks are due for a speedy review process which has further improved the quality of this work. The authors have addressed all the suggestions and comments. The description of the changes made is given as follows:
Reviewer #3:
1. Could you provide more details on the search strategy, including specific search terms, date ranges, and inclusion/exclusion criteria for the selected studies?
Response: Thanks for the suggestion. The required information has been updated in section 2.
2. While the review provides a comprehensive overview of MOF-based hydrogels for cancer treatment, it would benefit from a more critical evaluation of the studies' strengths and limitations.
Response: The strengths of individual case studies have been updated in Table 2 and the limitations of 3. MOF-based hydrogel have been updated in Section 5 “Challenges and future perspective”.
Some figures, particularly Figures 2-7, appear low-resolution. Please provide higher-quality images to enhance readability.
Response: All the figures have been updated and the high-resolution images have been shared with the journal.
4. Discuss any potential biases in the selected studies and how they might impact the interpretation of results.
Response: This review includes research articles related to MOF-based hydrogels (2017-2014) mainly used to treat cancer. We believe that this review will provide a comprehensive holistic view to the researchers for developing efficient MOF-based hydrogels with augmented anticancer effects enabling the effective management of cancer even without adverse effects.
5. Expand the discussion on the challenges and opportunities for translating MOF-based hydrogels from preclinical studies to clinical applications.
Response: Section 5. Challenges and future perspectives have been thoroughly modified.
6.Include a section that directly compares the efficacy and safety of MOF-based hydrogels with other existing cancer treatments or drug delivery systems.
Response: The content has been updated in Section 3.
7. Given the rapidly evolving nature of this field, please make sure that the review includes the most recent relevant studies (up to 2024).
Response: We have searched again the literature using different search engines and included the updated literature in the respective subsections.
8. Based on the gaps identified in the current literature, offer more specific recommendations for future research directions.
Response: Future perspectives have been updated in Section 5 and highlighted in red.
9. Consider adding a summary table that compares key features (e.g., synthesis method, cancer type, therapeutic mechanism) of the different MOF-based hydrogels discussed in the review.
Response: Table 3 (now Table 2) presents the summary of the literature discussed in the review.
10.The introduction could be enhanced by incorporating more detailed information. For example, the author can add more details using the following paper and referencing it:
https://www.sciencedirect.com/science/article/abs/pii/S1742706123006025
Response: Thanks for the suggestion. The introduction has been thoroughly revised and highlighted in red along with the proper citation of reference..
Thank you very much for your consideration.
Yours Sincerely,
Dr. Preeti Kush
Round 2
Reviewer 2 Report
Comments and Suggestions for Authors
After reviewing the revised manuscript, I have confirmed that all requested revisions have been appropriately addressed. The revised content meets the standards of the journal, and I believe it is now ready for publication. However, it has the following revised parts. They should be checked prior to the publication. Followings are recommended for the revision
Minor revision
1. page4,line144,145,149,151,152,153/page5,line167/page6,line182/table1,2/page21, line265/ page27, line475/page30, line560,563/page31, line605 Italicize in situ wherever they appear in these sections.
2. page 27, line 459, Italicize in- vivo
Author Response
To Date: 16.01.2025
Editorial Department of Gels
Dear Editor,
I am thankful for a positive response from your office on our manuscript (gels-3416303). Special thanks are due for a speedy review process which has further improved the quality of this work. The authors have addressed all the suggestions and comments. The description of the changes made is given as follows:
Reviewer #2:
Minor revision
1. page4,line144,145,149,151,152,153/page5,line167/page6,line182/table1,2/page21, line265/ page27, line475/page30, line560,563/page31, line605 Italicize in situ wherever they appear in these sections.
Response: All the corrections have been made and the text is highlighted in yellow.
2. page 27, line 459, Italicize in- vivo
Response: Thanks for the suggestion. The correction has been made and the text is highlighted in yellow.
Thank you very much for your consideration.
Yours Sincerely,
Dr. Preeti Kush
Reviewer 3 Report
Comments and Suggestions for Authors
The authors address all the comments, and the paper is in good shape for publication
Author Response
Thanks for kind reply.